# A computational model for structural dynamics and reconfiguration of DNA assemblies

Jae Young Lee ®[1], Heeyuen Koh[2] & Do-Nyun Kim ®[1,2,3,4] ✉

Recent advances in constructing a structured DNA assembly whose configuration can be dynamically changed in response to external stimuli have demanded the development of an efficient computational modeling approach to expedite its design process. Here, we present a computational framework capable of analyzing both equilibrium and non-equilibrium dynamics of structured DNA assemblies at the molecular level. The framework employs Langevin dynamics with structural and hydrodynamic finite element models that describe mechanical, electrostatic, base stacking, and hydrodynamic interactions. Equilibrium dynamic analysis for various problems confirms the solution accuracy at a near-atomic resolution, comparable to molecular dynamics simulations and experimental measurements. Furthermore, our model successfully simulates a long-time-scale close-to-open-to-close dynamic reconfiguration of the switch structure in response to changes in ion concentration. We expect that the proposed model will offer a versatile way of designing responsive and reconfigurable DNA machines.

DNA self-assembly, since its conception[1], has evolved into a versatile and robust approach to constructing structures with precise control over the shape and physical properties for various applications[2]. The development of DNA origami[3] has significantly advanced the synthesis and design strategies for structured DNA assemblies[4–6]. As a result, there has been a high demand for computational models of self-assembled DNA structures to investigate their folding mechanisms and underlying physics and to predict their shape and properties in the design phase prior to experimental synthesis. In particular, accurate and efficient analysis of the dynamic properties of the DNA structures has become crucial due to the rapidly growing need for structures that can dynamically change their conformation and hence functional properties in response to external stimuli.

The computational cost of all-atomic molecular dynamics (MD) simulations for analyzing origami-scale DNA structures, which could require millions to billions of degrees of freedom in an ionic solution[7–10], is considerably high. Therefore, many coarse-grained models have been proposed as efficient alternatives for their analysis. For example, oxDNA is a coarse-grained MD model that has been widely adopted for investigating the structural properties[11], self-assembly[12], and energy landscape of DNA structures[13]. A multi-resolution approach, mrDNA, is another coarse-grained dynamics model proposed to accelerate the prediction by linking different coarse-graining scales[14]. Alternatively, structural models have been developed by modeling molecular interactions between bases or base-pairs using beam and spring finite elements with equivalent mechanical properties. CanDo is the first structural model demonstrating the capability to quickly predict the shape and mechanical properties of lattice-based[15] and lattice-free DNA origami structures[16,17]. SNUPI is a more recent model achieving both accuracy and efficiency by employing a multiscale approach where the intrinsic geometric and mechanical properties of DNA motifs were systematically characterized using the MD simulations[18–20]. While these structural models can provide results more quickly than other coarse-grained models, they are limited to predicting the mean

[1]Institute of Advanced Machines and Design, Seoul National University, 1 Gwanak-ro, Gwanak-gu, Seoul 08826, Korea. [2]Soft Foundry Institute, Seoul National University, 1 Gwanak-ro, Gwanak-gu, Seoul 08826, Korea. [3]Department of Mechanical Engineering, Seoul National University, 1 Gwanak-ro, Seoul 08826, Korea. [4]Institute of Engineering Research, Seoul National University, 1 Gwanak-ro, Gwanak-gu, Seoul 08826, Korea. ✉e-mail: dnkim@snu.ac.kr

configuration with static stiffness values and equilibrium dynamic properties.

In this study, we developed a computational framework for the equilibrium and non-equilibrium dynamic analysis of DNA structures (Fig. 1a). This framework combined the structural model of SNUPI for DNA structures with a hydrodynamic model that considers the viscous effect and random force of an ionic solution. A model for the base stacking interaction was developed as well to simulate the reconfiguration of DNA structures in response to changes in ion concentration. The trajectory was calculated through a developed time-integration scheme for the Langevin equation, enabling us to rapidly obtain the dynamic trajectories of DNA structures with molecular-level precision. We verified the accuracy and efficiency of the proposed model by investigating various dynamic characteristics of representative DNA structures.

## Results

### Analysis framework

In the proposed framework, the dynamics of structured DNA assemblies in a solvent was described using Langevin dynamics simulations where the thermal effect of the surrounding solvent was modeled as an external random force so as to reduce the size of the system significantly (Supplementary Note 1). We started by assuming each base-pair as a rigid block with six (three translational and three rotational) degrees of freedom represented by a node and modeling the interactions between nodes using finite elements (Fig. 1b). In this way, a structured DNA assembly was converted into a set of structural and hydrodynamic finite element models.

In the structural model, we derived the internal force vector and mass matrix based on the formulation of SNUPI[18], characterizing the mechanical and electrostatic interactions of DNA helices at the

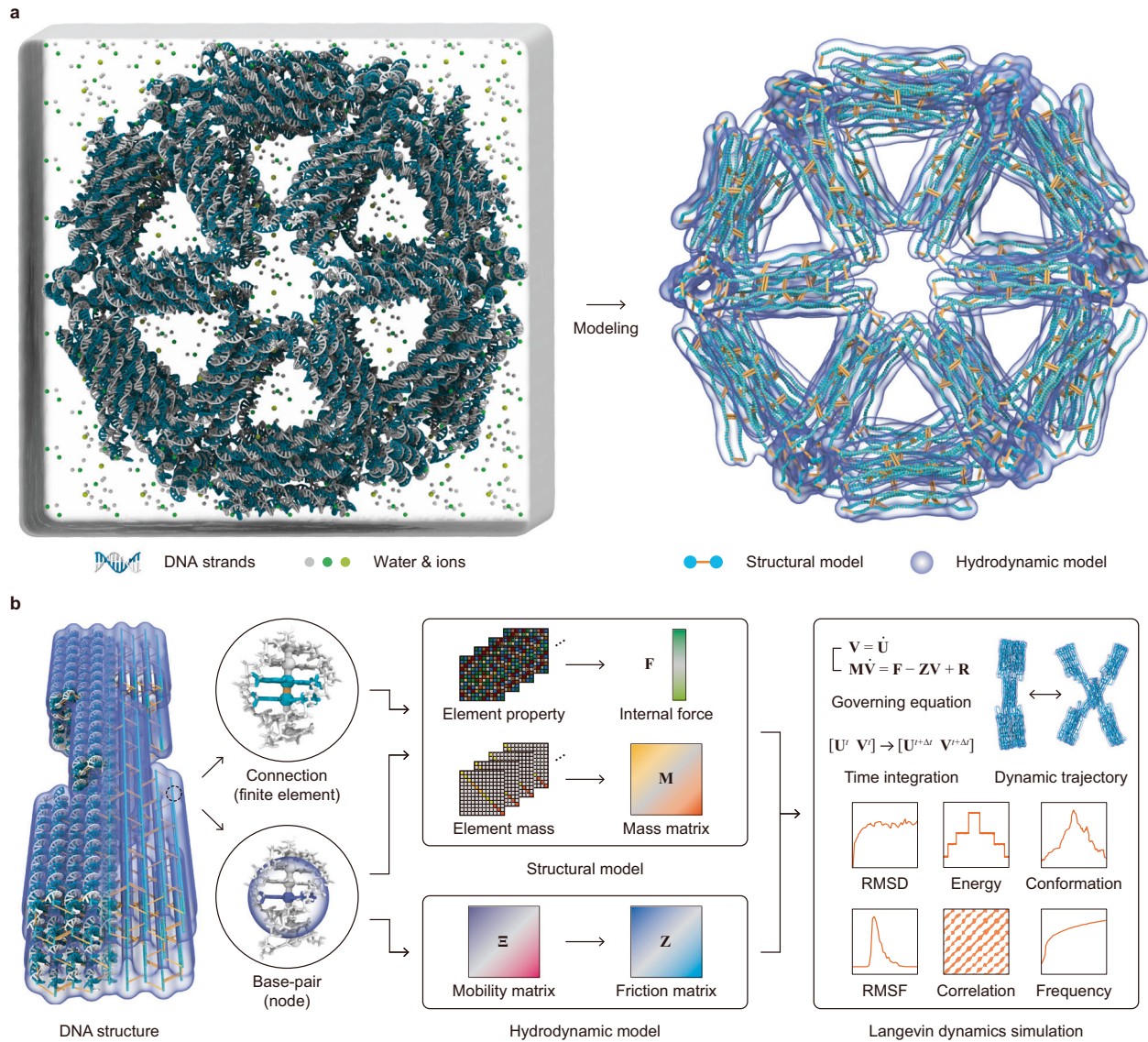

**Fig. 1 | Computational framework for the dynamic analysis of structured DNA assemblies. a** Schematic representation of the framework. The proposed approach combines a structural model for DNA structures based on mechanical and electrostatic forces with a hydrodynamic model that considers the viscous effect and random force of the ionic solution. **b** Overview of modeling. Each base-pair was defined as a node with three translational and three rotational degrees of freedom, and the connections between nodes were modeled as finite elements. The internal force vector was obtained from the coordinates of the nodes and the geometry and properties of structural and electrostatic connections. The mass matrix was generated using the information on the mass of base sequences and the structural connection. The friction matrix was constructed using the generalized Rotne-Prager-Yamakawa mobility matrix[21] to consider the effect of viscosity and random thermal force on the structure in the solvent. The dynamic trajectory was computed through a developed time-integrating scheme, allowing us to rapidly obtain trajectories of DNA structures with molecular-level precision.

molecular level (Supplementary Note 2). The internal force vector was constructed using the nodal coordinates and element properties of the finite element models in order to apply net mechanical and electrostatic forces. The mass matrix was generated using information on the mass of base sequences and the structural connection.

In the hydrodynamic model, we assumed the Stokes flow of viscous, linear, and incompressible fluids, considering the length scale of structured DNA assemblies. The generalized Rotne-Prager-Yamakawa mobility matrix[21] was employed to construct the friction matrix where six translational and rotational degrees of freedom for each node were considered as in the structural model (Supplementary Note 3). This mobility matrix accounted for the damping of a DNA structure with a complex shape through the superposition of identical spherical particles with a hydrodynamic radius assigned to each node. Then, the inverse of the mobility matrix provided the friction matrix under the linearity of the Stokes flow, quantifying the effect of the viscosity and random thermal force acting on the structure in the solvent.

The governing equations for Langevin dynamics were solved to obtain the trajectories of the DNA structure by using a direct time integration algorithm that we developed. Half-time stepping and Simpson's rule were introduced in the calculation of the internal force vector based on the Grønbech-Jensen Farago scheme[22] (Supplementary Note 4). Its application to representative linear problems (thermal diffusion in a flat potential and thermal harmonic oscillator) confirmed significantly improved numerical stability and performance compared to the original Grønbech-Jensen Farago scheme, enabling us to use a larger step size in time integration and hence to simulate more efficiently. To rapidly reach an equilibrium configuration, the dynamic analysis often began from a statically obtained configuration by minimizing the energy of structural and electrostatic potentials. The overall flow of the proposed analysis framework is described in Supplementary Fig. 1.

## Global shape

To validate the proposed analysis framework, we first investigated the mean shape of ten DNA wireframe structures in equilibrium. They included six triangular, two square, and two hexagonal structures, categorized into two types of edges: two-helix bundle (DX)[23] and 6-helix bundle (6HB)[24]. The dynamic simulations began from statically predicted energy-minimum configurations and were performed 500-ns-long simulations for them (Fig. 2). As expected, the structures with stiffer 6HB edges showed smaller means and variances in the root-mean-square deviation (RMSD) values than those with softer DX edges (Fig. 2a and Supplementary Table 1). The RMSD profiles further indicated that square and hexagonal structures with DX edges were flexible enough to reach another energy-minimum configuration more easily during the dynamic simulations, deviating slightly from the initial configuration obtained from static energy minimization (Supplementary Figs. 2 and 3).

Larger wireframe structures designed to fit within a circle of 25 nm radius maintained their planar shapes well (Fig. 2b). Those with DX edges were slightly distorted with the out-of-plane undulation due to their inherent flexibility. The predicted distributions of interior angles closely matched the experimentally measured angle distributions[24] (Fig. 2c). Nevertheless, the measured angle for the triangle with DX edges exhibited higher variation than the predicted one, potentially attributed to a high probability of structural fractures or fragments observed in micrographs[24] (Supplementary Table 2). We also calculated the out-of-plane angles of four small triangular structures, which are difficult to be quantified in experiments, for comparison with MD simulation results[24] (Fig. 2d, e). Our model could reproduce MD simulation results quite well, particularly for the structures with 6HB edges (Supplementary Table 3). Similar to larger wireframe structures, triangles with softer DX edges exhibited higher out-of-plane angles, as

also indicated by high root-mean-square fluctuation (RMSF) values at their vertices (Supplementary Fig. 4).

The overall predicted shapes of various wireframe structures were parallel to the reported cryo-EM data[25,26], although the structural edges were slightly more crooked than the experimental measurements (Fig. 2f). This was pronounced in the three-dimensional structures with DX edges (Supplementary Fig. 5) compared to the planar structures with 6HB edges (Supplementary Fig. 6), which indicates the higher rigidity of 6HB edges than DX ones. Moreover, structures with vertices connecting three or more edges through single-stranded DNA exhibited greater distortion attributable to the local stress when compared to the cryo-EM maps, suggesting a need for further investigation and modeling of complex junctions with multiple single-stranded DNAs.

In addition, we conducted simulations of transformable structures using modular dynamic units[27] (Supplementary Fig. 7), as well as reversible structures employing the transition between single-stranded and double-stranded DNA[28] (Supplementary Fig. 8). The structural transformations observed through dynamic simulations showed good agreement with previous reports, thereby validating the accuracy of the proposed model.

## Structural features at multiple levels

We further evaluated the performance of our model by analyzing the structural features of the pointer design at multiple levels whose high-resolution structure was determined experimentally using cryo-electron microscopy (cryo-EM)[29]. Its overall dimension and right-handed distortion about the helical axis could be accurately predicted with the model (Fig. 3a). The mean configuration obtained from the dynamic analysis was closer to the experimental structure with the RMSD of 12 Å than the one predicted by the static approach[18] resulting in the RMSD of 15 Å (Fig. 3b).

The principal component analysis (PCA) on the simulated trajectories (Supplementary Note 5) could successfully reveal the low-frequency, large-amplitude breathing motion[30] of the pointer structure (Fig. 3c). It appeared in the first mode and was coupled with structural rotation in a helical direction (Supplementary Fig. 9). The estimated breathing frequency was 2.35 GHz, which was lower than the structural vibrations observed in small DNA motifs (16 to 150 base-pairs) with 300 to 600 GHz[31,32] (Supplementary Table 4). This breathing motion could be predicted using the normal mode analysis (NMA) in a vacuum as well, but an unreasonably small frequency of 1.83 MHz was predicted (Fig. 3d). The mode shapes obtained from NMA were similar to those from PCA (Supplementary Fig. 10), but the natural frequencies were overestimated particularly for high-frequency modes, as the effect of solvent damping was not considered.

We then scrutinized the local geometry between successive base-pairs in the pointer structure. Six rigid-body parameters for base-pair steps in the 3DNA definition[33], including three translations (shift, slide, and rise) and three rotations (tilt, roll, and twist), were measured for all base-pairs in the structures obtained using the static analysis, dynamic analysis, and cryo-EM (Fig. 3e). The statically determined parameters were similar to the experimental ones on average, but they showed much narrower distributions. The distributions of these parameters obtained using the proposed dynamic analysis, on the other hand, were well matched with the experimental ones. Similar trends were observed for the parameters of interhelical crossovers (Fig. 3f). Unlike the sharper distributions of crossover parameters obtained in the static analysis, dynamically determined ones exhibited distributions similar to the experiment[29] and MD simulation[14]. Our model predicted smaller variances, particularly in the twist of crossovers (gamma), probably because our model assumed base-pairs remained intact but they could be broken at the crossover sites in reality or MD simulations. In the experimental cryo-EM structure[29], approximately 18% of base-pairs at crossover sites were broken, suggesting the softening of crossovers with deformation (Supplementary Fig. 11).

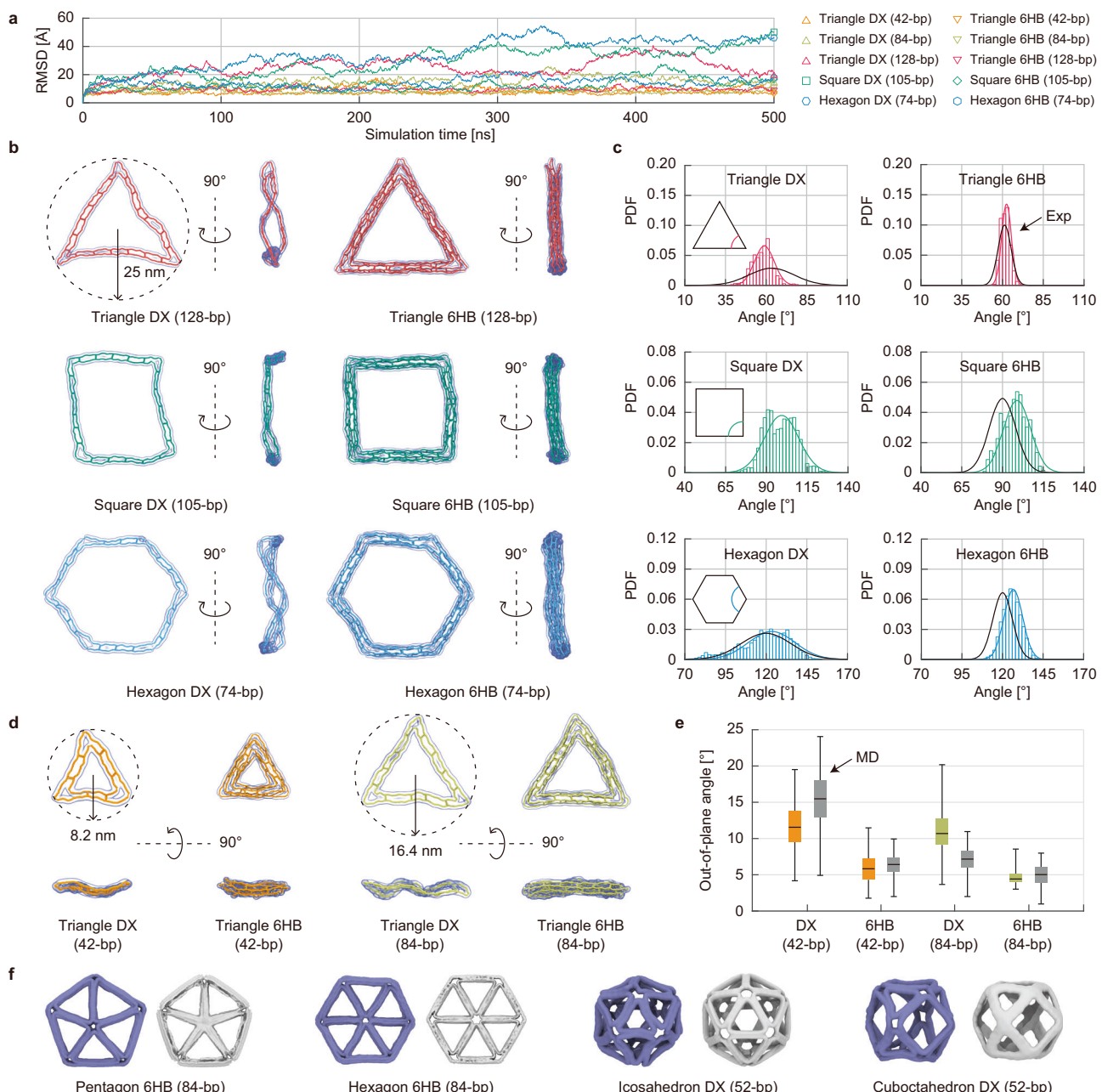

**Fig. 2 | Global shape of wireframe structures in equilibrium. a** Dynamic simulations of ten wireframe DNA structures. The graph indicates the root-mean-square deviation (RMSD) values over a 500-ns-long simulation for each structure. **b** Predicted shapes of six types of DNA structures. The triangular, square, and hexagonal structures with DX and 6HB edges were designed with a 25 nm radius. The images illustrate the top and side views of the structures at 500 ns. **c** Measured interior angles of vertices. The interior angle deviation for each structure was demonstrated in probability density distribution (PDF) and compared with the previous experiments[24]. **d** Predicted shapes of four triangular structures. The images show the top and bottom views of the structures. **e** Measured out-of-plane angles of vertices. The box plots of the out-of-plane angles were compared with the reported MD results[24] (sample size: 1,000). **f** Comparison of wireframe structures with experimental data. Predicted shapes (violet) and cryo-EM maps[25,26] (white) were demonstrated.

## Dynamic characteristics in equilibrium

We studied the model capability in predicting the equilibrium dynamic properties by analyzing the 12-helix-bundle (12HB) structure (Fig. 4a and Supplementary Fig. 12). We performed 60-ns-long simulations using the proposed method, as well as oxDNA[11] and mrDNA[14] models, and their results were compared with the MD[34] and NMA results[18]. The RMSD values converged within the first 20 ns for all simulations (Fig. 4b). The root-mean-square fluctuation (RMSF) amplitudes (Fig. 4c) and two (Pearson and generalized) correlation maps[35] at the base-pair level were evaluated using the final 20-ns-long trajectories

(Fig. 4d, e). For NMA, the lowest 200 normal modes were used to calculate these quantities.

All approaches could predict the thermal fluctuation amplitudes at the base-pair level quite accurately (Fig. 4c). The overlap coefficients calculated by $\vec{x}_{MD} \cdot \vec{x} / |\vec{x}_{MD}||\vec{x}|$, where $\vec{x}_{MD}$ and $\vec{x}$ represent vectorized RMSF values obtained from the MD and other methods, respectively, were 0.98 for the proposed approach, 0.97 for NMA, and 0.96 for oxDNA and mrDNA. The overall distribution of fluctuational amplitudes was similar among the methods although oxDNA predicted slightly larger amplitudes.

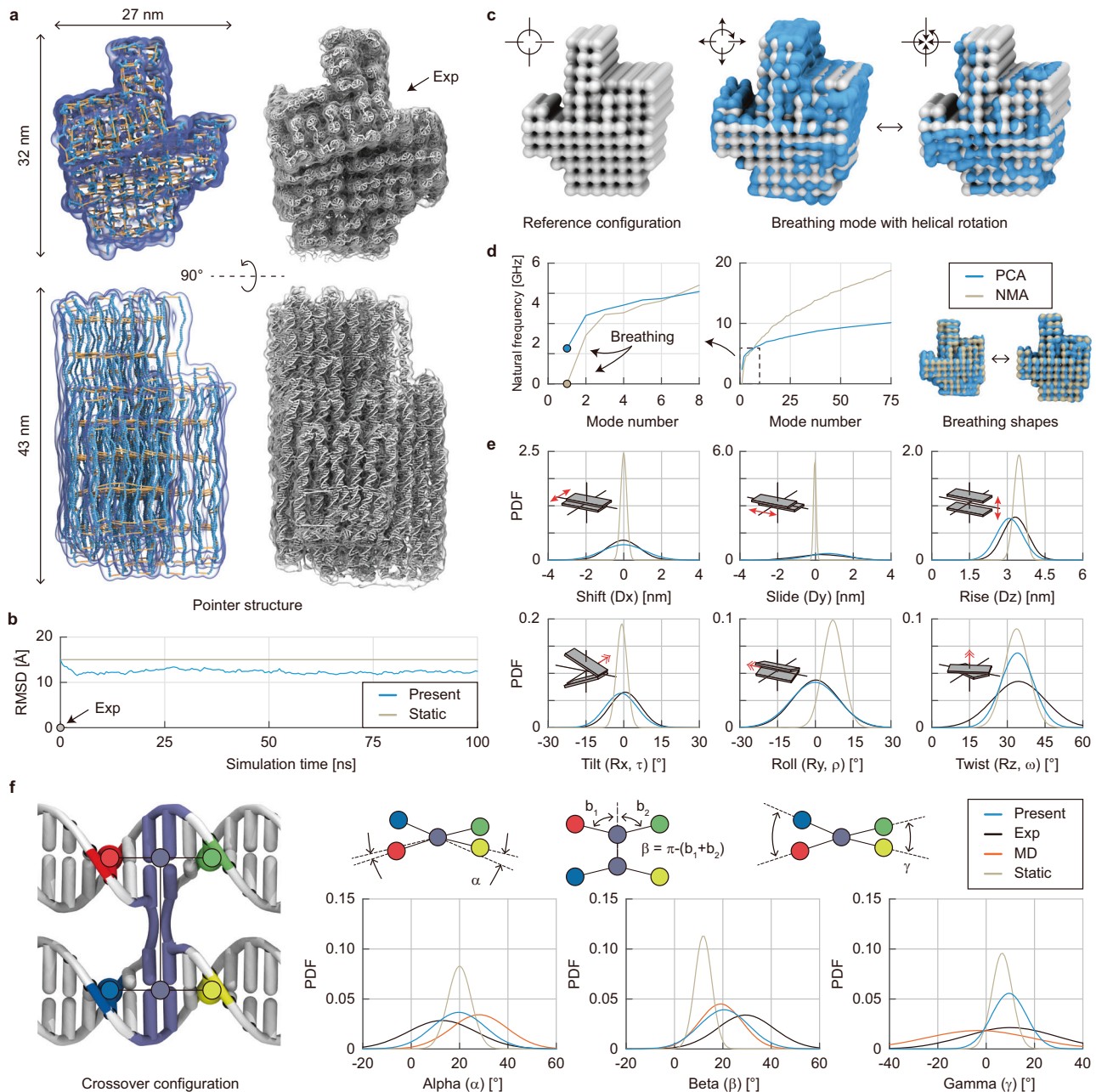

**Fig. 3 | Prediction of structural features at multiple levels. a** Comparison of simulated and experimental pointer structures. The simulated structure at the final 100 ns shows similar global dimensions to the cryo-EM structure[29]. The right-handed distortion about the helical axis was consistent with the design on the square lattice. **b** RMSD profile in the simulation. Starting from the statically predicted energy-minimum configuration with a RMSD of 15 Å, the simulation reached an equilibrium with an RMSD of 12 Å. **c** Breathing motion. Principal component analysis (PCA) was performed to observe the breathing motion of the pointer structure. The structural breathing orthogonal to the axis of constituent helices was captured, coupled with the rigid-body rotation of the structure. **d** Natural frequencies of the structure obtained using PCA and normal mode analysis (NMA). PCA estimated the natural frequency of 2.35 GHz for the breathing motion, while NMA provided a much smaller one of 1.83 MHz. The breathing shapes from PCA and NMA were demonstrated. **e** Geometric parameters of base-pair steps. They were calculated using the 3DNA definition[33]. The distributions of these parameters were matched well with the experimental ones[29], unlike the statically predicted values. **f** Crossover angles at interhelical connections. Three crossover angles were much closer to the experimental[29] and MD[14] results than the static predictions.

On the other hand, DNA structures exhibit a variety of conformational changes, which include complex motion between local base-pairs. The understanding of their correlated motions could be important to relate a DNA structure and function like proteins[35,36] or design functional structures. Therefore, we evaluated the accuracy in correlated motions between the proposed and other models and MD simulations. Notably, the proposed method predicted the correlation maps most similar to the MD results. It could reproduce the primary pattern of Pearson correlation coefficients in the map (upper triangular part of the map in Fig. 4d), which measure a linear, directional relationship of molecular motions based on the normalized covariance matrix of thermal fluctuations. In addition, our method captured longer-range correlations observed in the MD results better compared to other methods. The root-mean-square error (RMSE) of Pearson correlation coefficients for the proposed method with respect to MD was the smallest (0.186), followed by 0.212 for NMA and 0.213 for oxDNA. This directional correlation was not well captured by mrDNA with the

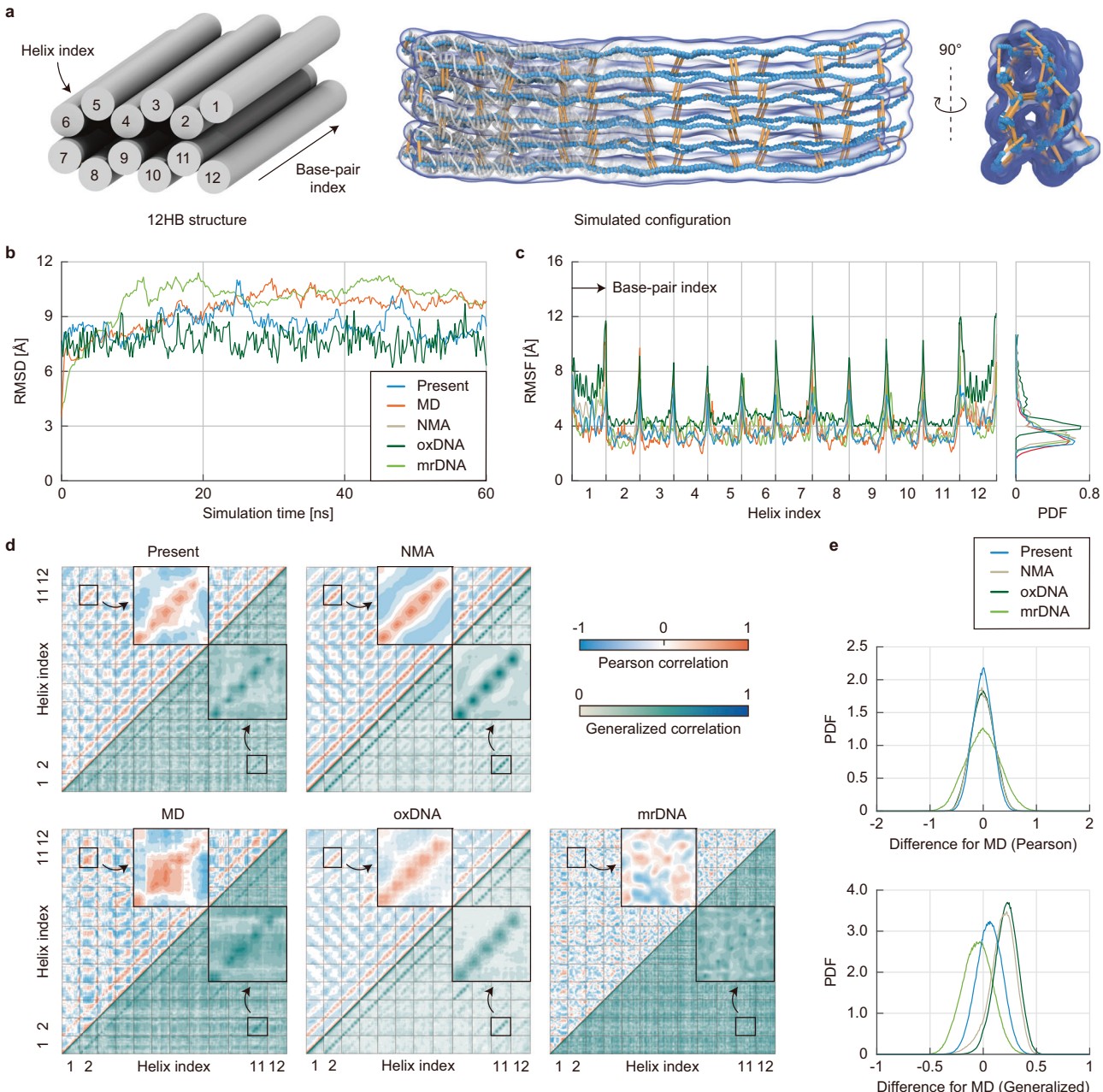

**Fig. 4 | Dynamic characteristics in equilibrium. a** Schematics of the 12-helix-bundle (12HB) structure. The helix index denotes the straight helices in a honeycomb lattice, and the base-pair index indicates the node in a helix from one end to the other. The configuration represents the final snapshot of the structure. **b** RMSD profiles in the simulation. We performed the analysis using the proposed method, oxDNA[11], and mrDNA[14] for the comparison with the MD results[34]. **c** Root-mean-square fluctuation (RMSF) amplitudes. For each approach, the RMSF profile and its normalized histogram as a probability density function (PDF) are shown.

**d** Correlation maps. Pearson and generalized correlations were calculated using the final 20-ns-long trajectories. The enlarged plots demonstrate the correlations between two adjacent helices connected by five crossover sites (helix index: 2 and 11). **e** Normalized histograms of the correlation maps. The vectorized difference in the Pearson and generalized correlation maps between MD and the other approaches was used to draw the probability density function. The difference value was calculated as $\vec{c}_{MD} - \vec{c}$ where $\vec{c}_{MD}$ and $\vec{c}$ represent vectorized correlation values obtained from MD and other methods, respectively.

biggest RMSE of 0.320. Similar results were obtained for the generalized correlation coefficients, which measure nonlinearly correlated motions based on the mutual information (lower triangular part of the map in Fig. 4d). Longer-range interactions could be better predicted by the proposed method with the smallest RMSE of 0.136. NMA and oxDNA tended to underestimate the generalized correlation overall, with strong peaks between nearby base-pairs. In contrast, mrDNA overestimated it with the biggest RMSE of 0.247.

## Ion responsive reconfiguration

To demonstrate the capability of the proposed approach, we aimed to analyze the ion-responsive reconfiguration of the switch design[37] (Fig. 5a). It consists of two arms pivoted at the center and joined by sixteen stacking bonds to form a closed configuration. Close-to-open reconfiguration can be triggered by lowering the salt concentration, which increases the repulsive electrostatic force between helices. To simulate this reconfiguration using the proposed method, it was crucial to characterize and model the base stacking interaction properly.

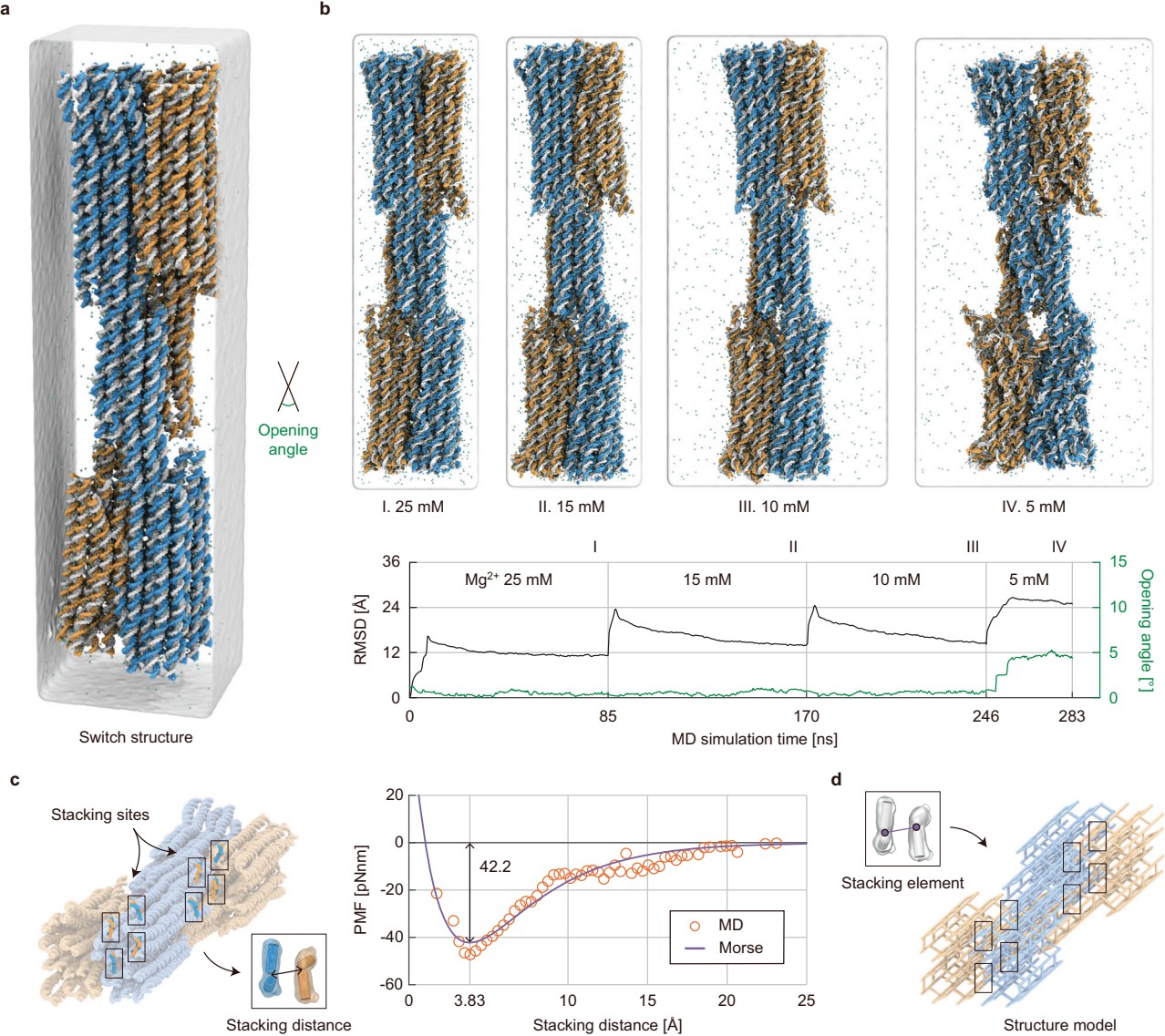

**Fig. 5 | All-atomic MD simulation of the switch structure. a** Switch structure. The two arms are joined by stacking bonds, holding a closed configuration. At low magnesium concentrations, an increase in repulsive electrostatic force within the structure can trigger the opening. The opening angle was defined as the angle between two arms. **b** MD simulation. The simulation started in the closed configuration. The maximum opening angle of 5° was smaller than 50° in the experiment[37]. **c** Characterizing and modeling the base stacking interaction. The potential of mean force (PMF) was obtained by analyzing the MD trajectories of the stacking distance between two base-pairs at sixteen stacking sites. The parameters of the Morse potential were adjusted to fit the MD results. The stacking free energy and effective region were estimated to be 42.2 pNnm and 25 Å, respectively. **d** Conversion of stacking interactions into structural finite elements. The stacking potential was converted into a finite element (stacking element) where the internal force exerted on two stacking nodes was generated by the gradient of stacking energy.

Toward this end, we performed an all-atom MD simulation of the switch design for 283 ns, gradually lowering the magnesium ion concentration from 25 mM in the closed configuration to 15, 10, and 5 mM (Supplementary Table 5). Due to the excessively high computational cost of the MD simulation, the maximum opening angle of 50° between two arms observed in the experiment[37] could not be reached, and only the initiation of the opening (up to the angle of 5°) was obtained (Fig. 5b). Nevertheless, we could observe highly dynamical interactions between two base-pairs at the sixteen stacking sites enabling us to model the stacking interaction. From the MD trajectories, the potential of mean force (PMF) was first obtained as a function of stacking distance (Fig. 5c, Supplementary Figs. 13 and 14). We employed the Morse potential to model the stacking interaction, including the effect of bond breaking, and its parameters were determined by fitting the PMF data (Supplementary Note 6). The stacking free energy and effective region were estimated to be 42.2 pNnm (6.08 kcal) and 25 Å, respectively, parallel to previous studies[38,39]. Finally, this stacking interaction model was implemented using a finite element (stacking element), which imposed the forces on two stacking nodes by calculating the gradient of stacking energy (Fig. 5d).

Using the model enriched with base stacking interactions, we performed a 21.4-μs-long Langevin dynamics simulation of the switch structure in order to simulate close-to-open-to-close dynamic reconfigurations (Fig. 6a). It began from the closed configuration at 25 mM of $Mg^{2+}$, and its conformational changes with respect to the salt concentration could be successfully predicted. The switch structure remained in its closed configuration until the ion concentration was decreased to 10 mM, but started to open at 5 mM, similar to the deformation tendency observed in our MD simulation. Its opening angle reached approximately 50° in agreement with the experiment

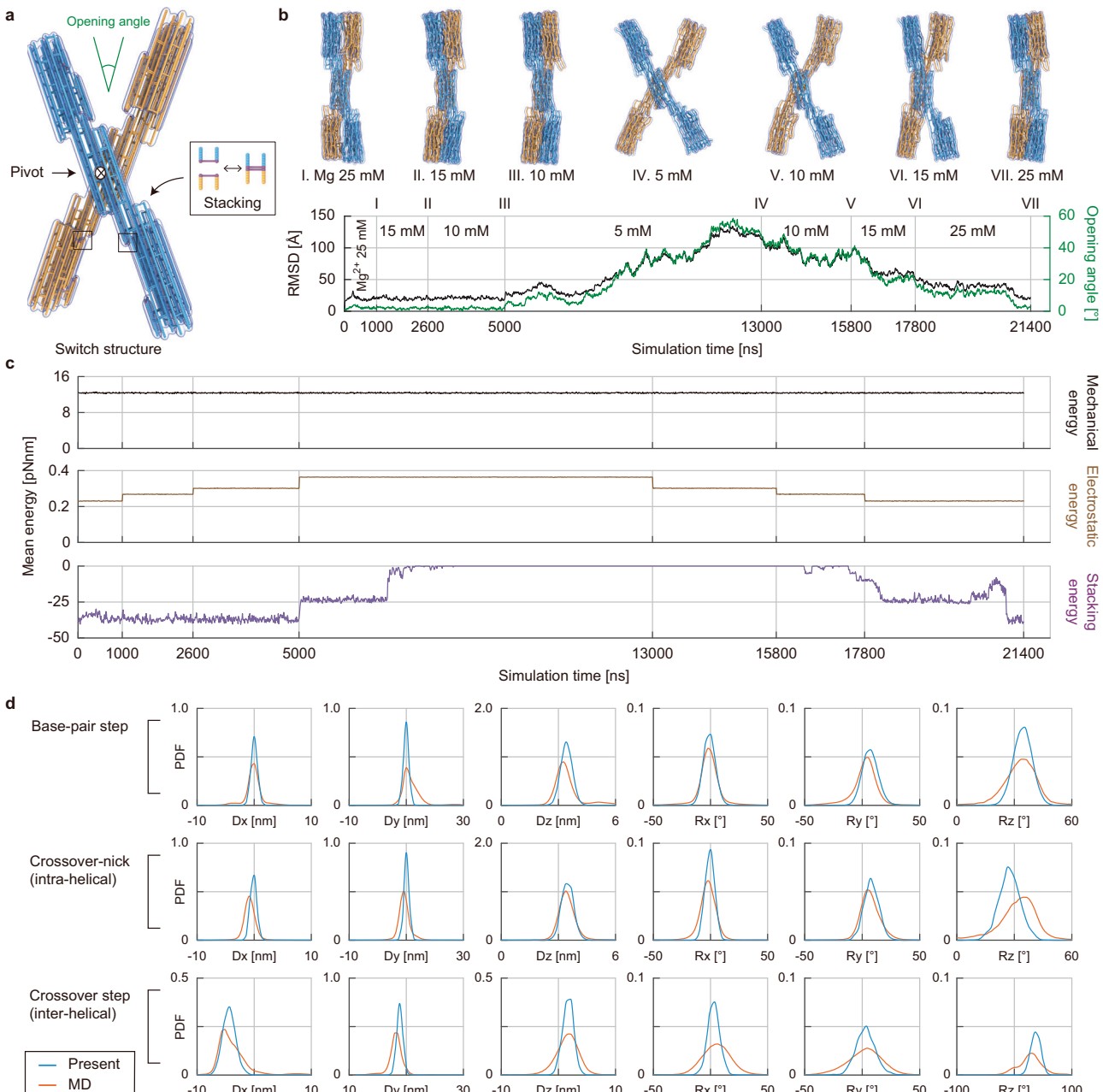

**Fig. 6 | Ion-responsive reconfiguration of the switch structure. a** The computational model of the switch structure. **b** Simulated reconfiguration with changes in ion concentration. The simulation started from the closed configuration. RMSD values (black) and opening angles (green) were collected during the simulation at different magnesium concentrations. The structure was widened swiftly to an opening angle of 50° at Mg²⁺ 5 mM while it was gradually closed as the concentration increased. **c** Energy trajectories. The mechanical energy remains constant since rigid-body motion mainly governs the reconfiguration, while the base stacking energy shows a time delay with respect to the electrostatic energy. **d** Geometric parameters. The geometric information of the proposed model was compared with the MD results.

when 5 mM of Mg²⁺ was maintained and came back to 0° of the closed state as the ion concentration increased again to 25 mM (Fig. 6b). Since the structural reconfiguration of the switch design was almost rigid-body motion, the mechanical energy in the structure did not vary significantly (Fig. 6c). According to the stacking and electrostatic energy profiles, it took some time for the bases at the sixteen stacking sites to be fully unstacked and restacked in response to the change in the ion concentration (Supplementary Fig. 15). During the opening process, the six bases at one end of the structure were unstacked first almost immediately after the ion concentration was lowered to 5 mM, followed by the unstacking of the remaining ten bases after 2 μs

(Supplementary Fig. 16). During closing, the latter ten bases were stacked first at 25 mM, followed by the stacking of the other six bases.

Furthermore, we compared the geometric parameters at the base-pair level from the molecular trajectories obtained using the MD simulation and the proposed method. The overall distributions of six geometric parameters obtained using our method were closely matched with those from the MD simulation for all connections between successive base-pairs, including base-pair steps and inter-helical and intra-helical steps at crossover sites (Fig. 6d). Our model predicted narrower distributions of the rotational parameters particularly for base-pairs at crossover sites as base-pair breakages were not modeled.

Similar results were also obtained when characterizing these parameters at a specific ion concentration (Supplementary Figs. 17–20). RMSF (Supplementary Figs. 21–24) and correlation maps (Supplementary Figs. 25–28) showed good agreement with MD results except for the case of 5 mM of $Mg^{2+}$ in which the open state could not be reached in MD simulations. These results demonstrated the capability of the proposed model to accurately capture the long-time-scale dynamics and structural information at molecular resolution. It is also noteworthy that the proposed method could generate dynamic trajectories with a sampling speed of 36 ns per hour using a single workstation while that of 0.22 ns per hour was only possible for the MD simulation even using a supercomputer with 24,000 CPU cores (Supplementary Table 6).

## Discussion

We demonstrated that the proposed computational modeling approach could accurately infer the dynamic properties of structured DNA assemblies. The efficient analysis was achieved by describing mechanical, hydrodynamic, electrostatic, and base stacking interactions using finite element models in the Langevin dynamics framework. Notably, the ion-mediated transient response of the switch structure for reconfiguration could be simulated successfully using the proposed method, which was inaccessible with the MD simulation. This work can pave the way for investigating complex structural dynamics and enabling the rational design of dynamic molecular machines such as DNA-based rotors[40,41]. Our model also has potential for further development toward models for diffusion and hydrodynamics of structured DNA assemblies by introducing an external flow field[42,43]. While the current model was tested for dynamic analysis of monomeric structures, it would be possible to extend its capability to simulate the assembly and disassembly process of supramolecular polymeric structures[44,45]. Also, the reconfiguration process controlled by other types of stimuli such as chemo-mechanical reconfiguration by DNA binding molecules[46,47] could be simulated if a model for conformational changes in DNA double helices by their binders would be incorporated into the proposed computational framework.

## Methods

### Dynamic analysis framework

We considered the Langevin dynamics equation (Supplementary Note 1) as $\mathbf{M}\dot{\mathbf{V}}^t = \mathbf{F}^t - \mathbf{Z}^t\mathbf{V}^t + \mathbf{R}^t$ and $\dot{\mathbf{U}}^t = \mathbf{V}^t$, where $\mathbf{U}^t$ and $\mathbf{V}^t$ are the coordinate and velocity vectors at a simulation time ($t$), respectively, connected by the time derivatives, $\mathbf{M}$ is a time-independent diagonal mass matrix, $\mathbf{F}^t$ and $\mathbf{Z}^t$ are the internal force vector and friction matrix, respectively, and $\mathbf{R}^t$ is a random force vector. Based on the dissipation-fluctuation theorem, the random force vector was assumed to be Gaussian distributed with the statistical properties as $\langle\mathbf{R}^t\rangle = 0$ and $\langle\mathbf{R}^t\mathbf{R}^{\tau\top}\rangle = 2k_B T\mathbf{Z}^t\delta(t-\tau)$, where $k_B$ is the Boltzmann constant, and $T$ is the absolute temperature of the heat bath as 300 K.

In the structural model, to construct the mass matrix ($\mathbf{M}$) and internal force vector ($\mathbf{F}^t$), we employed SNUPI[18] as the multiscale approach to construct the finite element assembly of DNA structures using molecular-level properties (Supplementary Note 2). The structural connections of a DNA structure were categorized into base-pair steps (intrahelical ones), crossover steps (interhelical ones), and end-to-end connection of single-stranded DNA. We used their intrinsic geometry and mechanical properties characterized through molecular dynamics simulations[18,19,48]. The mass matrix was constructed from the structure information and sequence-dependent mass values[49].

In the hydrodynamic model, we constructed the friction matrix ($\mathbf{Z}^t$) using the generalized Rotne-Prager-Yamakawa mobility matrix ($\mathbf{\Xi}^t$)[21], which provided six degrees of freedom for each node, similar to the structural model (Supplementary Note 3). Assuming the Stokes flow surrounding DNA structures, the friction matrix ($\mathbf{Z}^t$) can be calculated by inverting the mobility matrix as $\mathbf{Z}^t = (\mathbf{\Xi}^t)^{-1}$. The positive definiteness of the mobility matrix was guaranteed. The hydrodynamic radius of each node was assumed to be $\sigma = 1.1$ nm, and the dynamic viscosity of water was set to $\eta = 890$ μN s/m² at 300 K from the previous studies[50,51].

The temporal trajectory of the coordinate and velocity vectors was numerically updated by performing time integration as $[\mathbf{U}^t, \mathbf{V}^t] \rightarrow [\mathbf{U}^{t+\Delta t}, \mathbf{V}^{t+\Delta t}]$, where $\Delta t$ is the time interval (Supplementary Note 4 and Supplementary Figs. 29 and 30). We modified the Grønbech-Jensen Farago scheme[22] by introducing half-time stepping and Simpson's rule in calculating internal force. In general, the time interval was set to 5 ps, and the internal force vector was updated at every time step, while the friction matrix, which had a little change between time steps, was updated at intervals of 1000 to 10000 relatively slowly. For time scaling, the time interval or simulated time were used to real values without introducing a scaling factor as like all-atomic MD. Initial velocities of nodes with mass $m_i$ were randomly generated using Gaussian distribution with zero mean and deviation of $k_B T/m_i$.

Using the dynamic trajectory, principal component analysis (PCA) was performed (Supplementary Note 5). Assuming the quasi-harmonic energy[52], the principal modes are derived using the fluctuation matrix as $\boldsymbol{\sigma} = \langle(\mathbf{x}-\langle\mathbf{x}\rangle)(\mathbf{x}-\langle\mathbf{x}\rangle)^\top\rangle$, where $\mathbf{x}$ is the position trajectory of nodes and the angle bracket represents the time average. The mass-weighted fluctuation matrix ($\boldsymbol{\Sigma}$) is then calculated using the mass matrix as $\boldsymbol{\Sigma} = \mathbf{M}^{1/2}\boldsymbol{\sigma}\mathbf{M}^{1/2}$. The eigenmodes of the structure are derived by performing normal mode analysis as $\boldsymbol{\Sigma}\boldsymbol{\Phi} = \boldsymbol{\Phi}\boldsymbol{\Lambda}$, where $\boldsymbol{\Phi}$ and $\boldsymbol{\Lambda}$ are the set of eigenvectors and eigenvalues, respectively. The natural frequency is finally computed as $\omega_i = (k_B T/\Lambda_i)^{1/2}$ and mode shapes is given by $\Delta\mathbf{x}_i = (\mathbf{M}^{1/2})^{-1}\phi_i$.

### Molecular dynamics simulations

We performed all-atom molecular dynamics (MD) simulations using the program NAMD[53] with the CHARMM36 force-field for nucleic acids[54]. The DNA structures were explicitly solvated using the TIP3P water model[55] and ionized using sodium ($Na^+$), magnesium ($Mg^{2+}$), and chloride (Cl⁻) molecules in the cubic cell with periodic boundary conditions. The short-range electrostatic and Lennard-Jones potentials employed a cutoff of 10 Å, and its switching function was active above 8 Å. The long-range electrostatic interactions were calculated using the particle-mesh-Ewald method[56] with a 1 Å grid spacing.

For the switch structure[37], an idealized atomic structure was generated from the caDNAno[57] design using the export function of SNUPI[18]. In the ionization process, magnesium–hexahydrate complexes were randomly placed near the structure due to slow diffusion[58]. To prevent the dissociation of these complexes, harmonic restraints were placed between the magnesium atom and the oxygen atoms, with the spring constant of 10 kcal/mol Å². The initial concentration of $Mg^{2+}$ was set to 25 mM. The atomic system was prepared using the visual molecular dynamics (VMD)[59].

After constructing the initial system of the switch structure, the atomic potential energy was minimized using 5,000 static steps. In the dynamic simulations, we employed a time step of 2 fs and maintained pressure and temperature using the Nosé–Hoover Langevin piston and Langevin thermostat, respectively, for all non-hydrogen atoms with a damping constant of 1 ps⁻¹. During the thermalization process, the temperature was gradually increased by 1 K every 500 steps until reaching 300 K, while the pressure was maintained at 1 bar. All production runs were performed in the isobaric–isothermal (NPT) ensemble at 1 bar and 300 K. After the production simulation with 25 mM $Mg^{2+}$, the magnesium concentration was gradually reduced to 15, 10, and 5 mM, and the system size was increased to over ten million atoms.

We considered the stacking interaction between distant two base-pairs (Supplementary Note 6). The Morse potential function was used

to model the stacked and unstacked base-pairs[60,61] given by $\Pi_{SK}(r) = \varepsilon[[1 - \exp(-a(r - r_0))]^2 - \varepsilon$, where $r$ is the distance, $\varepsilon$ represents the energy parameter for dissociation of stacking, a denotes the shape parameter, and $r_0$ is the equilibrium distance. To fit the model parameters, we used the potential of mean force (PMF) obtained from the MD simulations as $\Pi_{PMF}(r) = -k_B T \log[g(r)]$, where $g(r)$ represents the radial distribution function of the stacking distances. The resulting parameters from the MD simulations were $\varepsilon = 42.79$ pNnm, $a = 2.668$ nm$^{-1}$, and $r_0 = 0.3742$ nm.

## OxDNA simulations

To simulate the 12HB structure, we employed the standalone GPU-enabled oxDNA2 model[60]. The ionic solution condition was set to Na$^+$ 0.5 M, which was reasonable to mimic the experimental conditions of DNA origami with magnesium[11]. The simulations were performed for 60 ns with a time step of 15 fs using a Langevin thermostat at 300 K. The velocity and angular momentum of each node were refreshed every 103 time steps.

## MrDNA simulations

To examine the fluctuation and correlation of 12HB structure in equilibrium, we performed the simulation using the GPU-enabled mrDNA model[14], which introduced a multi-resolution approach to accelerate dynamic simulations. Starting from an idealized configuration, the structure was simulated at a coarse resolution (five base-pairs per bead) for 20 ns using a time step of 200 fs. Subsequently, a successive simulation was performed at a fine resolution (two beads per base-pair) for 40 ns using a time step of 50 fs.

## Reporting summary

Further information on research design is available in the Nature Portfolio Reporting Summary linked to this article.

## Data availability

All data are included in the article, the supplementary information, and the source data file. Source data are provided with this paper.

## Code availability

The proposed framework is provided through SNUPI (Structured NUcleic acids Programming Interface)[62], which is available at https://github.com/SSDL-SNU/SNUPI.

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

## Acknowledgements

This research was supported by the National Convergence Research of Scientific Challenges through the National Research Foundation of Korea (NRF) funded by Ministry of Science and ICT (NRF-2020M3F7A1094299), the Bio & Medical Technology Development Program of the National Research Foundation of Korea (NRF) funded by the Korean government (MSIT) (NRF-2022M3E5F1018465), and the National Supercomputing Center with supercomputing resources including technical support (KSC-2021-CHA-0025). J.Y.L. acknowledges the support by the National Research Foundation of Korea (NRF) funded by the Korea government (MSIT) (NRF-2021R1C1C2003554).

## Author contributions

J.Y.L. and D.-N.K. designed the study. J.Y.L. developed the approach and analyzed the data. H.K. commented on and examined the approach. J.Y.L. and D.-N.K. wrote and revised the manuscript.

## Competing interests

The authors declare no competing interests.
