## [Peer Review File · Nature Communications]

A computational model for structural dynamics and reconfiguration of DNA assembliesReviewer #1 (Remarks to the Author):

This interesting manuscript described an improved version of a previously described method (SNUPI) for characterization of the structural ensemble adopted by self-assembled DNA objects. The key new developments include a hydrodynamic description of friction and an explicit model for end-stacking interactions. The authors demonstrate the ability of their improved method to capture the structural ensembles explored by all-atom MD simulations and experiment and to simulate a conformational transition in a large DNA origami object triggered by an ion concentration change.

Overall, this is a nice study that pushes the field forward. Addressing the following will improve the manuscript.

It's a little surprising that the authors present a new integration approach in a paper that primarily describes a model developed for a specific simulation domain. Does the new integration scheme confer any advantages compared to other established integration schemes such as leapfrog or the Leimkuhler/Mathews geodesic integration scheme? Of course, an in-depth discussion could be reserved for another manuscript.

Looking at the supplementary figures, it appears that PCA has been performed without aligning the trajectory to a reference configuration, so that the top modes are (mostly) global translations and rotations. Would it not be better to neglect those global configurational changes so that the internal dynamics of the nanostructure can be appreciated? The same comment applies for the NMA.

It would be nice if the authors could explain their motivation for placing so much emphasis on the correlation analysis. Why are these quantities of interest? After all, the comparison is made to the all-atom description which also has its shortcomings regarding the timescale sampled and force field imperfections.

The authors should add a statement somewhere explaining if one should take literally the time scales reported for the simulations (as is the case for the all-atom MD method) or if one should assume a certain scaling factor to convert the quoted simulation duration or mode frequencies into real-world time scales (as is the case for the majority of the coarse-grained models).

The repertoire of structures simulated using the improved model is on a smaller side if compared to the previous studies on the same subject. Applying the method to a more diverse set of structures could broaden the appeal of the study.

Finally, the manuscript leaves an impression that the present method can accurately describe hydrodynamics. Can it? If it can, please simulate a single DNA helix and/or a long six-helix bundle in a laminar flow and quantify the effect of the flow on the average conformation and conformational fluctuations. If such capability is presently beyond the scope of the method, please state that, for example, in the discussion section of the manuscript and adjust the writing of the manuscript accordingly.

Reviewer #2 (Remarks to the Author):

The authors present a powerful coarse-grained computational tool for simulating long time-scale conformational dynamics of structured DNA assemblies made from DNA origami. In particular, the model for the first time enables the prediction of large-scale conformational rearrangements such as ionic-strength-mediated caliper opening and closing that are inaccessible to all atom molecular dynamics simulations due to the time-scales involved. They further show their model improves accuracy of conformational dynamics predictions considerably over competing approaches such as mrDNA. They demonstrate the capabilities of their approach on a diversity of DNA origami assemblies including DX- and 6HB-based wireframe assemblies as well as a brick-like Pointer

assembly, comparing with all-atom simulations and cryoEM structural data. The presented tool should prove particularly valuable for the design of functional DNA assemblies that are of high molecular weight and contain large conformational rearrangements, such as in molecular sensing and robotics applications.

The manuscript is well written with rigorous methods and controls presented, both computational and experimental. My only question for revision is why the authors chose not to incorporate comparison of their modeling predictions with cryoEM data of wireframe assemblies (either 6HB 2D/3D or DX-based 3D), which would offer additional concrete validation of the modeling predictions (e.g., subtle aspects of twist and bend along wireframe edges depending on wireframe geometry).

Response to Reviewers

Manuscript ID: NCOMMS-23-27111A

Title: "A computational model for structural dynamics and reconfiguration of DNA assemblies"

Authors: Jae Young Lee, Heeyuen Koh, and Do-Nyun Kim

We thank the Reviewers very much for the thoughtful comments. The details of the point-by-point response to the Reviewer's comments are as follows.

Reviewer 1

This interesting manuscript described an improved version of a previously described method (SNUPI) for characterization of the structural ensemble adopted by self-assembled DNA objects. The key new developments include a hydrodynamic description of friction and an explicit model for end-stacking interactions. The authors demonstrate the ability of their improved method to capture the structural ensembles explored by all-atom MD simulations and experiment and to simulate a conformational transition in a large DNA origami object triggered by an ion concentration change.

Overall, this is a nice study that pushes the field forward. Addressing the following will improve the manuscript.

- It's a little surprising that the authors present a new integration approach in a paper that primarily describes a model developed for a specific simulation domain. Does the new integration scheme confer any advantages compared to other established integration schemes such as leapfrog or the Leimkuhler/Mathews geodesic integration scheme? Of course, an in-depth discussion could be reserved for another manuscript.
- Looking at the supplementary figures, it appears that PCA has been performed without aligning the trajectory to a reference configuration, so that the top modes are (mostly) global translations and rotations. Would it not be better to neglect those global configurational changes so that the internal dynamics of the nanostructure can be appreciated? The same comment applies for the NMA.
- It would be nice if the authors could explain their motivation for placing so much emphasis on the correlation analysis. Why are these quantities of interest? After all, the comparison is made to the all-atom description which also has its shortcomings regarding the timescale sampled and force field imperfections.
- The authors should add a statement somewhere explaining if one should take literally the time scales reported for the simulations (as is the case for the all-atom MD method) or if one should assume a certain scaling factor to convert the quoted simulation duration or mode frequencies into real-world time scales (as is the case for the majority of the coarse-grained models).
- The repertoire of structures simulated using the improved model is on a smaller side if compared to the previous studies on the same subject. Applying the method to a more diverse set of structures could broaden the appeal of the study.
- Finally, the manuscript leaves an impression that the present method can accurately describe hydrodynamics. Can it? If it can, please simulate a single DNA helix and/or a long six-helix bundle in a laminar flow and quantify the effect of the flow on the average conformation and conformational fluctuations. If such capability is presently beyond the scope of the method, please state that, for example, in the discussion section of the manuscript and adjust the writing of the manuscript accordingly.

Comment 1.1

It's a little surprising that the authors present a new integration approach in a paper that primarily describes a model developed for a specific simulation domain. Does the new integration scheme confer any advantages compared to other established integration schemes such as leapfrog or the Leimkuhler/Mathews geodesic integration scheme? Of course, an in-depth discussion could be reserved for another manuscript.

Response 1.1

To our knowledge, the proposed approach for time integration has higher numerical accuracy compared to the original GJF or leapfrog algorithms, allowing the use of large time intervals, even though requiring additional information (half-time coordinates, auxiliary matrices, etc.) to predict the configuration at the next time step ($t + \Delta t$) from a specific time step (t).

In addition, our algorithm is efficient in matrix operations to simulate Langevin dynamics based on finite element modeling. Some reported algorithms such as Bussi and Parrinello's (Bussi, G. and Parrinello, M., 2007. *Accurate sampling using Langevin dynamics. Physical Review E*, 75(5), p.056707.) or Jensen's (Jensen, L.F.G. and Grønbech-Jensen, N., 2019. *Accurate configurational and kinetic statistics in discrete-time Langevin systems. Molecular Physics*, 117(18), pp.2511-2526.) could require exponential and square root operations of large matrices so need large computing memory and time, while the proposed algorithm only requires addition, subtraction, and inverse matrix calculations like the original GJF algorithm.

We thought that a comparative and systematic investigation with various established integration algorithms would be better considered in further studies. In addition, the geodesic integration method (Leimkuhler and Matthews's) was for Langevin simulations with constraints, which could be implemented for efficient time integration in future work.

We added a sentence in the 'Supplementary Note 4' section as follows.

(...) Note that the sampling of the configurational space is the main objective of the simulations of DNA systems, which is an overdamped case ($\gamma/2\Omega > 30$, region C in Fig. N1). Therefore, considering the stability region, numerical tests, and robust configurational sampling properties, we could choose a large time step (Δt) using the developed algorithm for a DNA system. **In future studies, the proposed algorithm could be improved more efficiently by employing constrained Langevin dynamics¹⁷.**

The also added the reference as follows.

- Leimkuhler, B. and Matthews, C., 2016. Efficient molecular dynamics using geodesic integration and solvent-solute splitting. *Proceedings of the Royal Society A: Mathematical, Physical and Engineering Sciences*, 472(2189), p.20160138.

Comment 1.2

Looking at the supplementary figures, it appears that PCA has been performed without aligning the trajectory to a reference configuration, so that the top modes are (mostly) global translations and rotations. Would it not be better to neglect those global configurational changes so that the internal dynamics of the nanostructure can be appreciated? The same comment applies for the NMA.

Response 1.2

Yes, following the reviewer's comment, we neglect the modes for the global configurational changes (rigid-body motions) in PCA and NMA. We first modified Fig. 3d to show the natural frequency of internal modes as follows.

- From:

- To:

We revised the paragraph in the 'Structural features at multiple levels' section of the Results part as follows.

- From: (...) The principal component analysis (PCA) on the simulated trajectories (Supplementary Note 5) could successfully reveal the low-frequency, large-amplitude breathing motion³⁰ of the pointer structure (Fig. 3c). **It appeared in the sixth mode and was coupled with a rigid-body rotation of the structure** (Supplementary Fig. 9). The estimated breathing frequency was 2.35 GHz, which was lower than the structural vibrations observed in small DNA motifs (16 to 150 base-pairs) with 300 to 600 GHz^{31,32} (Supplementary Table 4). This breathing motion could be predicted using the normal mode analysis (NMA) in a vacuum as well, but an unreasonably small frequency of 1.83 MHz was predicted (Fig. 3d). The mode shapes obtained from NMA were similar to those from PCA (Supplementary Fig. 10), but the natural frequencies were overestimated particularly for high-frequency modes, as the effect of solvent damping was not considered. (...)
- To: (...) The principal component analysis (PCA) on the simulated trajectories (Supplementary Note 5) could successfully reveal the low-frequency, large-amplitude breathing motion³⁰ of the pointer structure (Fig. 3c). **It appeared in the first mode and was coupled with structural rotation in a helical direction** (Supplementary Fig. 9). The estimated breathing frequency was 2.35 GHz, which was lower than the structural vibrations observed in small DNA motifs (16 to 150 base-pairs) with 300 to 600 GHz^{31,32} (Supplementary Table 4). This breathing motion could be predicted using the normal mode analysis (NMA) in a vacuum as well, but an unreasonably small frequency of 1.83 MHz was predicted (Fig. 3d). The mode shapes obtained from NMA were similar to those from PCA (Supplementary Fig. 10), but the natural frequencies were overestimated particularly for high-frequency modes, as the effect of solvent damping was not considered. (...)

We also modified Supplementary Figs. 9 and 10, to show the mode shapes of nine internal modes as follows.

Comment 1.3

It would be nice if the authors could explain their motivation for placing so much emphasis on the correlation analysis. Why are these quantities of interest? After all, the comparison is made to the all-atom description which also has its shortcomings regarding the timescale sampled and force field imperfections.

Response 1.3

DNA structures exhibit a variety of structural changes involving complex motions between local base-pairs in solution. MD simulations can provide detailed and accurate information on correlated motions but are limited in the time scale as the reviewer commented. Therefore, we assessed the accuracy of base-pair-level correlated motions through the proposed method and previously reported models. We think that the analysis and evaluation of the correlated motions could be important to understand the structural characteristics and functions like proteins or design functional DNA structures.

Following the reviewer's comment, we added the introductory sentences to the 'Dynamic characteristics in equilibrium' section of the Result part as follows.

- From: (...) The overall distribution of fluctuational amplitudes was similar among the methods although oxDNA predicted slightly larger amplitudes. Notably, the proposed method predicted the correlation maps most similar to the MD results. It could reproduce the primary pattern of Pearson correlation coefficients in the map (upper triangular part of the map in Fig. 4d), which measure a linear, directional relationship of molecular motions based on the normalized covariance matrix of thermal fluctuations. (...)
- To: (...) The overall distribution of fluctuational amplitudes was similar among the methods although oxDNA predicted slightly larger amplitudes.

On the other hand, DNA structures exhibit a variety of conformational changes, which include complex motion between local base-pairs. The understanding of their correlated motions could be important to relate the DNA structure and function like proteins^{35,36} or design functional structures. Therefore, we evaluated the accuracy in correlated motions between the present and other models and MD simulations. Notably, the proposed method predicted the correlation maps most similar to the MD results. It could reproduce the primary pattern of Pearson correlation coefficients in the map (upper triangular part of the map in Fig. 4d), which measure a linear, directional relationship of molecular motions based on the normalized covariance matrix of thermal fluctuations. (...)

We also added a reference for the correlated motion in the manuscript as follows.

- Xu, D., Meisburger, S.P. and Ando, N., 2021. Correlated motions in structural biology. *Biochemistry*, 60(30), pp.2331-2340.

Comment 1.4

The authors should add a statement somewhere explaining if one should take literally the time scales reported for the simulations (as is the case for the all-atom MD method) or if one should assume a certain scaling factor to convert the quoted simulation duration or mode frequencies into real-world time scales (as is the case for the majority of the coarse-grained models).

Response 1.4

Thank you for the careful comment. We did not introduce a scale factor for time scale, so they can be used as it is. Therefore, we added a statement in the ‘Dynamic analysis framework’ section of the Methods part as follows.

- From: (...) In general, the time interval was set to 5 ps, and the internal force vector was updated at every time step, while the friction matrix, which had a little change between time steps, was updated at intervals of 1000 to 10000 relatively slowly. The time scales of simulations were used as it is since no scaling factor was introduced. Initial velocities of nodes with mass m_i were randomly generated using Gaussian distribution with zero mean and deviation of $k_B T/m_i$. (...)
- To: (...) In general, the time interval was set to 5 ps, and the internal force vector was updated at every time step, while the friction matrix, which had a little change between time steps, was updated at intervals of 1000 to 10000 relatively slowly. Initial velocities of nodes with mass m_i were randomly generated using Gaussian distribution with zero mean and deviation of $k_B T/m_i$. (...)

Comment 1.5

The repertoire of structures simulated using the improved model is on a smaller side if compared to the previous studies on the same subject. Applying the method to a more diverse set of structures could broaden the appeal of the study.

Response 1.5

Following the reviewer's comment, we predicted and analyzed more experimentally validated DNA structures.

First, we predicted eight DNA origami structures made of small modular dynamic units and conducted a comparative analysis of their shapes with experiments (Wang *et al.*). From the same initial configuration, the dynamic simulations began and gradually produced different shapes, such as elongated, bent, and twisted configurations, by programming the combination of modular dynamic units. In the converged configurations, the structural shapes showed good agreement with experimental measurements. Please see Supplementary Fig. 7 below.

Secondly, we introduced a reversible DNA origami structure using the transition between single-stranded DNA (ssDNA) and double-stranded DNA (dsDNA) and validated with experimental results (Gür *et al.*). The initially opened configuration, stabilized by rigid dsDNA molecules, was changed into a closed configuration by the entropic force of ssDNA. Our dynamic simulation successfully showed the reversible transition of open-closed-open states and corresponding shapes. Please see Supplementary Fig. 8 below.

Finally, we predicted seven 2D and six 3D wireframe structures with DX and 6HB edges and systematically analyzed their shapes and local differences with cryo-EM maps. Please see Comment 2.1.

We then added a paragraph in the 'Global shape' section of the Results part as follows.

(...) In addition, we conducted simulations of transformable structures using modular dynamic units²⁷ (Supplementary Fig. 7), as well as reversible structures employing the transition between single-stranded and double-stranded DNA²⁸ (Supplementary Fig. 8). The structural transformations observed through dynamic simulations showed good agreement with previous reports, thereby validating the accuracy of the proposed model.

The two references for new structures were added in the manuscript as follows.

- Wang, D., Yu, L., Huang, C.M., Arya, G., Chang, S. and Ke, Y., 2021. Programmable transformations of DNA origami made of small modular dynamic units. *Journal of the American Chemical Society*, 143(5), pp.2256-2263.
- Gür, F.N., Kempter, S., Schueder, F., Sikeler, C., Urban, M.J., Jungmann, R., Nickels, P.C. and Liedl, T., 2021. Double- to single-strand transition induces forces and motion in DNA origami nanostructures. *Advanced Materials*, 33(37), p.2101986.

We added Supplementary Figs. 7 and 8, to show the dynamic simulations of added DNA structures with the experimental results as follows.

Supplementary Fig. 7. Simulation of DNA structures with modular dynamic units

The DNA origami structures with modular dynamic units (modular expandable origami, MEO)²⁸ were predicted and compared with experiments. There were two 26-bp-long (gray) and 36-bp-long (orange) dynamic units, and different shapes of structures were programmed by controlling their combination. Since the structures are highly deformable, the static analysis was not converged. Instead, we performed dynamic simulations that began from the same initial structure (MEO-26). The length of MEO-26, MEO-L2, MEO-L3, and MEO-36 and the bending angle of MEO-C1, MEO-C2, and MEO-C2 were compared with experimental distribution (mean and std).

Supplementary Fig. 8. Dynamic simulations of reversible DNA structures

The reversible DNA origami structure²⁹ using the transition between single-stranded DNA (ssDNA) and double-stranded DNA (dsDNA) was simulated to show the structural transition. By changing the mechanical properties of helices from dsDNA to ssDNA in the supporting parts (orange), the opened configuration gradually deformed into a closed one. The closed structure was re-opened by assigning the ssDNA properties again.

Comment 1.6

Finally, the manuscript leaves an impression that the present method can accurately describe hydrodynamics. Can it? If it can, please simulate a single DNA helix and/or a long six-helix bundle in a laminar flow and quantify the effect of the flow on the average conformation and conformational fluctuations. If such capability is presently beyond the scope of the method, please state that, for example, in the discussion section of the manuscript and adjust the writing of the manuscript accordingly.

Response 1.6

Thank you for the interesting idea. Referring to the previous papers by Jendrejack *et al.* (Jendrejack, Richard M., Juan J. de Pablo, and Michael D. Graham. "Stochastic simulations of DNA in flow: Dynamics and the effects of hydrodynamic interactions." *The Journal of chemical physics* 116, no. 17 (2002): 7752-7759.) and Shaqfeh (Shaqfeh, Eric S.G. "The dynamics of single-molecule DNA in flow." *Journal of Non-Newtonian Fluid Mechanics* 130, no. 1 (2005): 1-28.), the velocity profile in a laminar flow can be considered in a Langevin dynamics simulation. This could enable us to predict the flow-dependent behavior of DNA helices or structures. However, the integration of the flow effect into the current approach requires a more careful consideration than simply introducing external velocity terms to our formulation, so this modeling is beyond the scope of this study.

Therefore, we included a statement in the 'Discussion' section as follows.

(...) This work can pave the way for investigating complex structural dynamics and enabling the rational design of dynamic molecular machines such as DNA-based rotors^{40,41}. Our model also has potential for further development toward models for diffusion and hydrodynamics of structured DNA assemblies by introducing an external flow field^{42,43}. While the current model was tested for dynamic analysis of monomeric structures, it would be possible to extend its capability to simulate the assembly and disassembly process of supramolecular polymeric structures^{44,45}. Also, the reconfiguration process controlled by other types of stimuli such as chemo-mechanical reconfiguration by DNA binding molecules^{46,47} could be simulated if a model for conformational changes in DNA double helices by their binders would be incorporated into the proposed computational framework.

The two references were added in the manuscript as follows.

- Jendrejack, R.M., de Pablo, J.J. and Graham, M.D., 2002. Stochastic simulations of DNA in flow: Dynamics and the effects of hydrodynamic interactions. *The Journal of chemical physics*, 116(17), pp.7752-7759.
- Shaqfeh, E.S., 2005. The dynamics of single-molecule DNA in flow. *Journal of Non-Newtonian Fluid Mechanics*, 130(1), pp.1-28.

Reviewer 2

The authors present a powerful coarse-grained computational tool for simulating long time-scale conformational dynamics of structured DNA assemblies made from DNA origami. In particular, the model for the first time enables the prediction of large-scale conformational rearrangements such as ionic-strength-mediated caliper opening and closing that are inaccessible to all atom molecular dynamics simulations due to the time-scales involved. They further show their model improves accuracy of conformational dynamics predictions considerably over competing approaches such as mrDNA. They demonstrate the capabilities of their approach on a diversity of DNA origami assemblies including DX- and 6HB-based wireframe assemblies as well as a brick-like Pointer assembly, comparing with all-atom simulations and cryoEM structural data. The presented tool should prove particularly valuable for the design of functional DNA assemblies that are of high molecular weight and contain large conformational rearrangements, such as in molecular sensing and robotics applications.

The manuscript is well written with rigorous methods and controls presented, both computational and experimental.

- My only question for revision is why the authors chose not to incorporate comparison of their modeling predictions with cryoEM data of wireframe assemblies (either 6HB 2D/3D or DX-based 3D), which would offer additional concrete validation of the modeling predictions (e.g., subtle aspects of twist and bend along wireframe edges depending on wireframe geometry).

Comment 2.1

My only question for revision is why the authors chose not to incorporate comparison of their modeling predictions with cryo-EM data of wireframe assemblies (either 6HB 2D/3D or DX-based 3D), which would offer additional concrete validation of the modeling predictions (e.g., subtle aspects of twist and bend along wireframe edges depending on wireframe geometry).

Response 2.1

Following the reviewer's comment, we further employed the cryo-EM data (EMDB) of six 3D wireframe structures with DX edges (Veneziano *et al.*) and seven 2D wireframe structures with 6HB edges (Wang *et al.*). We simulated the new structures for 500 ns and confirmed the convergence of RMSD curves. Their global and local shapes of predicted structures were then compared with corresponding cryo-EM maps.

Overall, the shapes of the predicted wireframe structures showed good agreement with the reconstructed cryo-EM maps. In the case of the 3D structures with DX edges, a slightly larger local distortion was observed than that of the 2D structures with 6HB edges, but the overall shapes were maintained well. In particular, in the 2D structure with vertices connecting three or more edges, the out-of-plane deformation was observed compared to cryo-EM maps, which could be related to the local stress generated at the vertices between single-stranded DNAs (for example, Pentagon with 84-bp-long 6HB edges and Hexagon with 84-bp-long 6HB edges. Please see the Supplementary Figs. 5 and 6 below). This deformation could be because the current modeling of single-stranded DNA (ssDNA) represents the mechanical behavior of independent strands. This encourages investigation of the mechanical effects of junctions of multiple ssDNAs, such as vertices in wireframe structures in further studies.

We added a paragraph in the 'Global shape' section of the Results part as follows.

(...) The overall predicted shapes of various wireframe structures were parallel to the reported cryo-EM data^{25,26}, although the structural edges were slightly more crooked than the experimental measurements (Fig. 2f). This was pronounced in the three-dimensional structures with DX edges (Supplementary Fig. 5) compared to the planar structures with 6HB edges (Supplementary Fig. 6), which indicates the higher rigidity of 6HB edges than DX ones. Moreover, structures with vertices connecting three or more edges through single-stranded DNA exhibited greater distortion attributable to the local stress when compared to the cryo-EM maps, suggesting a need for further investigation and modeling of complex junctions with multiple single-stranded DNAs. (...)

The corresponding two references for the reported cryo-EM data were added in the manuscript as follows.

- Veneziano, R., Ratanalert, S., Zhang, K., Zhang, F., Yan, H., Chiu, W. and Bathe, M., 2016. Designer nanoscale DNA assemblies programmed from the top down. *Science*, 352(6293), pp.1534-1534.
- Wang, X., Li, S., Jun, H., John, T., Zhang, K., Fowler, H., Doye, J.P., Chiu, W. and Bathe, M., 2022. Planar 2D wireframe DNA origami. *Science advances*, 8(20), p.eabn0039.

We illustrated the comparison of representative four 2D and 3D structures in the Fig. 2f as follows.

We also added Supplementary Figs. 5 and 6, to show the 2D and 3D shapes of all predicted wireframe structures with corresponding cryo-EM maps as follows.

Supplementary Fig. 6. Comparison of DNA wireframe structures (2D) with cryo-EM data

The seven planar DNA wireframe structures (6HB) were compared with cryo-EM maps²⁷. For each structure, the 500-ns-long dynamic trajectory was converted into density map (violet) using VMD²⁶ and illustrated along with the corresponding experimental maps (white).

Reviewer #1 (Remarks to the Author):

The authors have satisfactorily addressed all of the comments from the previous round of review. A few minor comments on the revised manuscript:

1. Consider revising the statement added in Response 1.4 since it doesn't read well without the context of the comment.

2. In the new SI Fig. 8, it isn't clear how the dsDNA/ssDNA switching region is modeled; in the figure the orange regions representing dsDNA in the first and last panel appear to be perfectly straight, and so not modeled in the usual way. Clarification would be nice.

Reviewer #2 (Remarks to the Author):

The authors have addressed my comments so I'm happy to endorse publication of their nice study.

Response to Reviewers

Manuscript ID: NCOMMS-23-27111A

Title: "A computational model for structural dynamics and reconfiguration of DNA assemblies"

Authors: Jae Young Lee, Heeyuen Koh, and Do-Nyun Kim

We thank the Reviewers very much again. The details of the point-by-point response to the Reviewer's comments are as follows.

Reviewer 1

The authors have satisfactorily addressed all of the comments from the previous round of review. A few minor comments on the revised manuscript:

- Consider revising the statement added in Response 1.4 since it doesn't read well without the context of the comment.
- In the new SI Fig. 8, it isn't clear how the dsDNA/ssDNA switching region is modeled; in the figure the orange regions representing dsDNA in the first and last panel appear to be perfectly straight, and so not modeled in the usual way. Clarification would be nice.

Comment 1.1

Consider revising the statement added in Response 1.4 since it doesn't read well without the context of the comment.

Response 1.1

Thank you for the kind comment. Following the reviewer's comment, we revised the statement in the 'Dynamic analysis framework' section of the Methods part as follows.

- From: (...) In general, the time interval was set to 5 ps, and the internal force vector was updated at every time step, while the friction matrix, which had a little change between time steps, was updated at intervals of 1000 to 10000 relatively slowly. **The time scales of simulations were used as it is since no scaling factor was introduced.** Initial velocities of nodes with mass m_i were randomly generated using Gaussian distribution with zero mean and deviation of $k_B T/m_i$. (...)
- To: (...) In general, the time interval was set to 5 ps, and the internal force vector was updated at every time step, while the friction matrix, which had a little change between time steps, was updated at intervals of 1000 to 10000 relatively slowly. **For time scaling, the time interval or simulation time was used to real value without introducing a scaling factor as like all-atomic MD.** Initial velocities of nodes with mass m_i were randomly generated using Gaussian distribution with zero mean and deviation of $k_B T/m_i$. (...)

Comment 1.2

In the new SI Fig. 8, it isn't clear how the dsDNA/ssDNA switching region is modeled; in the figure the orange regions representing dsDNA in the first and last panel appear to be perfectly straight, and so not modeled in the usual way. Clarification would be nice.

Response 1.2

Yes, following the reviewer's comment, we clarified the legend of the Supplementary Fig. 8 to describe the simulation details as follows.

- From: The reversible DNA origami structure²⁹ using the transition between single-stranded DNA (ssDNA) and double-stranded DNA (dsDNA) was simulated to show the structural transition. By changing the mechanical properties of helices from dsDNA to ssDNA in the supporting parts (orange), the opened configuration gradually deformed into a closed one. The closed structure was re-opened by assigning the ssDNA properties again.
- To: Structural reconfiguration of the reversible DNA origami structures²⁹ was simulated using transitions between single-stranded DNA (ssDNA) and double-stranded DNA (dsDNA). The mechanical properties of the finite elements in the transition region (orange) were changed as dsDNA-ssDNA-dsDNA to simulate the open-closed-open reconfiguration. Initially, the geometric and mechanical properties of dsDNA were assigned to ssDNA in the transition region and the structure was maintained in an open configuration. After the RMSD curve of the open structure converged, the structure was gradually transformed into a closed configuration by assigning ssDNA properties to the transition region. Finally, the structure was reopened by reassigning dsDNA properties to the transition region.

Reviewer 2

The authors have addressed my comments so I'm happy to endorse publication of their nice study.

Response

We thank you of all careful comments very much.